# Endocrine Disruptors in Food, Estrobolome and Breast Cancer

**DOI:** 10.3390/jcm12093158

**Published:** 2023-04-27

**Authors:** Alessio Filippone, Cristina Rossi, Maria Maddalena Rossi, Annalisa Di Micco, Claudia Maggiore, Luana Forcina, Maria Natale, Lara Costantini, Nicolò Merendino, Alba Di Leone, Gianluca Franceschini, Riccardo Masetti, Stefano Magno

**Affiliations:** 1Center for Integrative Oncology, Fondazione Policlinico Universitario A. Gemelli IRCCS, 00168 Rome, Italy; cristina.rossi1@guest.policlinicogemelli.it (C.R.); mariamaddalena.rossi1@policlinicogemelli.it (M.M.R.); annalisa.dimicco@guest.policlinicogemelli.it (A.D.M.); claudia.maggiore@guest.policlinicogemelli.it (C.M.); luana.forcina@guest.policlinicogemelli.it (L.F.); stefano.magno@policlinicogemelli.it (S.M.); 2Breast Cancer Center, Fondazione Policlinico Universitario A. Gemelli IRCCS, 00168 Rome, Italy; maria.natale@policlinicogemelli.it (M.N.); alba.dileone@policlinicogemelli.it (A.D.L.); gianluca.franceschini@policlinicogemelli.it (G.F.); riccardo.masetti@policlinicogemelli.it (R.M.); 3Department of Ecological and Biological Sciences (DEB), Tuscia University, Largo dell’Università snc, 01100 Viterbo, Italy; lara.cost@unitus.it (L.C.); merendin@unitus.it (N.M.); 4Women’s Health Department, Università Cattolica del Sacro Cuore, 00168 Rome, Italy

**Keywords:** microbiome, endocrine disruptors, estrobolome, personalized medicine, oncobiotic

## Abstract

The microbiota is now recognized as one of the major players in human health and diseases, including cancer. Regarding breast cancer (BC), a clear link between microbiota and oncogenesis still needs to be confirmed. Yet, part of the bacterial gene mass inside the gut, constituting the so called “estrobolome”, influences sexual hormonal balance and, since the increased exposure to estrogens is associated with an increased risk, may impact on the onset, progression, and treatment of hormonal dependent cancers (which account for more than 70% of all BCs). The hormonal dependent BCs are also affected by environmental and dietary endocrine disruptors and phytoestrogens which interact with microbiota in a bidirectional way: on the one side disruptors can alter the composition and functions of the estrobolome, ad on the other the gut microbiota influences the metabolism of endocrine active food components. This review highlights the current evidence about the complex interplay between endocrine disruptors, phytoestrogens, microbiome, and BC, within the frames of a new “oncobiotic” perspective.

## 1. Introduction

Breast cancer (BC) is currently one of the most prevalent cancers, with an estimated number of 2.3 million new cases worldwide [1]. It represents the fifth most common cause of cancer-related deaths [2].

BC incidence is expected to increase further, particularly in low- income countries, due to the westernization of lifestyles (e.g., lack of physical activity and poor diet), and improved cancer detection [3]. Current projections indicate that by 2030, the number of new cases diagnosed will reach 2.7 million annually [4].

The World Health Organization (WHO) distinguishes at least 18 different histological BC types among a wide spectrum of tumors featuring different morphologies, molecular characteristics, and clinical behaviors [5]. Invasive BC can be categorized into molecular subtypes based on mRNA gene expression levels independently of histological subtypes. In 2000, Perou et al. identified four molecular subtypes from microarray gene expression data: Luminal, HER2-enriched, Basal-like, and Normal Breast-like [6]; further studies allowed to divide the Luminal group into two subgroups (Luminal A and B) [7,8,9,10,11].

Luminal A tumors are characterized by the presence of estrogen-receptor (ER) and/or progesterone-receptor (PR) and absence of HER2. This subtype [12,13] is associated to a low expression of genes related to cell proliferation and shows a better prognosis, compared to Luminal B tumors, which are ER positive but may be PR negative and/or HER2 positive.

Overall, 80% and 65% of patients are diagnosed with BC positive for estrogen receptor (ER) and progesterone receptor (PR), respectively [9].

A new classification has recently been proposed for HER 2 tumors with a score of 1+ or 2+ without amplification by the ISH method (in situ hybridization); these are nicknamed HER 2 low breast cancer and account for more than half of all breast cancer cases.

On the basis of the latest studies, it has been seen that this subcategory of tumors could benefit from new anti-HER 2 drugs. However, we are far from being able to define HER 2 low tumors as a separate clinical entity with its prognosis and specific features [14].

Validation of techniques to identify HER2 heterogeneity in order to effectively treat tumors with non-uniform HER2 expression is needed [15].

BC is a multifactorial disease, and several genetic and environmental aspects are recognized as risk factors for its onset and progression [16]. Among them, age, and modifiable factors such as obesity, type II diabetes, sedentary habits, alcohol, radiation, hormonal replacement therapy, and periodontal disease have direct implications on gut microbiota composition, so that recent studies have highlighted the association between microbial alterations and those risk factors for BC, through metabolic and immunitary pathways, hormonal balance, and cancer microenvironment [17,18,19].

Regarding the sexual hormonal balance, estrogens, and endocrine active compounds play a role in shaping the gut microbiome, potentially impacting the clinical management of hormone-dependent cancers [20].

## 2. Endocrine Disruptors, Phytoestrogens and Breast Cancer

An Endocrine Disruptor (ED) is defined by the U.S. Environmental Protection Agency (EPA) as “an exogenous agent that interferes with synthesis, secretion, transport, metabolism, binding action, or elimination of natural blood-borne hormones that are present in the body and are responsible for homeostasis, reproduction, and developmental process” [21].

Both estrogens and EDs, binding to estrogen receptors, elicit downstream gene activation and trigger intracellular signalling cascades [22] in a variety of tissues, thus affecting reproductive health and hormonal dependent cancers risk [23,24,25].

Endocrine disruptors are a group of highly heterogeneous molecules, grossly divided into synthetic and natural compounds (phytoestrogens).

### 2.1. Synthetic Endocrine Disruptors

The synthetic chemicals with endocrine activities have multiple uses, such as industrial solvents/lubricants (polychlorinated biphenyls (PCBs), polybrominated biphenyls (PBBs)), plastics (bisphenol A (BPA)), plasticizers (phthalates), pesticides (methoxychlor, chlorpyrifos, dichlorodiphenyltrichloroethane (DDT)), fungicides (vinclozolin), pharmaceutical agents (diethylstilbestrol (DES)) and heavy metals such as cadmium [25,26].

The most common pathways of exposure to EDs are by inhalation, food intake, transplacental and skin contact [25,27,28]. By these means, EDs enter the food chain and accumulate in animal tissues up to humans mainly in adipose tissue, since most of EDs are highly lipophilic [29,30,31].

The mechanisms of action of EDs include a variety of possible pathways involved in endocrine and reproductive systems: via nuclear receptors, nonnuclear steroid hormone receptors (e.g., membrane estrogen receptors (ERs)), nonsteroid receptors (e.g., neurotransmitter receptors such as serotonin, dopamine, norepinephrine), orphan receptors [e.g., aryl hydrocarbon receptor (AhR)], enzymatic pathways involved in steroid biosynthesis and/or metabolism [25].

Another mechanism is the aromatase up-regulation (e.g., phenolic EDs) and increased estradiol biosynthesis, which is linked to ER-positive breast cancer cell proliferation in vitro [32].

Furthermore, an epigenetic action, such as DNA methylation and/or acetylation and histone modifications, may be involved in mechanisms related to endocrine disruption [33,34,35].

The exposure to EDs has been related to multiple diseases, such as diabetes, metabolic syndrome, obesity, cardiovascular and neurological disorders [29,30,31,32,33,34,35,36,37]. Some EDs such as bisphenol A (BPA), dichlorodiphenyltrichloroethane (DDT) and polychlorinated biphenyls (PCBs) are also associated with infertility and cancer [29,30,31,32,33,34,35,36,37,38,39,40].

According to the International Agency for Research on Cancer (IARC) classification, some of the EDs (BPA, DDT and PCBs) have key characteristics of human carcinogens, since they can alter cell proliferation, cell death or nutrient supply; are genotoxic; have immunosuppressive activity; induce epigenetic alterations, oxidative stress and chronic inflammation [39]. In addition, BPA by interacting with the estrogen receptor-α (ERα), induces cell proliferation and reduces apoptosis rate, affecting the prognosis of BC patients [40,41,42].

A growing number of studies have investigated the correlations between EDs and BC onset and progression [43]. Breast tissue is particularly susceptible to carcinogenic effects during the third trimester of the first pregnancy, and prolonged exposure to low levels of EDs [44,45,46] can raise the risk of developing cancer in the following years [47,48].

Some pesticides, including DDT, dichloro-diphenyl-dichloroethylene (DDE), aldrin, and lindane, have been linked in pre- and post-menopausal women to a higher risk of BC [49,50], either estrogen receptor-positive (-hexachlorocyclohexane and Pentachlorothioanisole) [51] or HER2-positive tumors (DDT) [52,53,54]. Among the heavy metals, cadmium was positively associated with BC [55,56]. 

Interestingly, women with an altered body composition and an excess of fat mass have shown a greater likelihood of BC after exposure to PCB [57], due to the lipophilic nature of these molecules.

Some EDs, such as Bisphenol S (BPS), are also involved in enhancing the progression and the metastatic spread of BC cells, by inducing tumor proliferation and epithelial-mesenchymal transition [58,59]. The Interplay between endocrine disruptors and microbiota with potential drivers of BC are summarized in Table 1.

### 2.2. Phytoestrogens

Due to their chemical structures and/or activities similar to 17-estradiol (E2) [38,67,68], some plant-derived polyphenolic non-steroidal substances, defined phytoestrogens, are classified as endocrine disruptors, with both potentially favorable (reduced risk of osteoporosis, heart disease, and menopausal symptoms) and harmful health consequences [69,70].

In epidemiological studies, Asian populations who consume on average much more soy products than Western populations, have lower rates of hormone-dependent breast and endometrial cancers [71] and a lower incidence of menopausal symptoms and osteoporosis. Soy is the main dietary source of isoflavones. Isoflavones have a chemical structure similar to the human hormone oestrogen. However, they bind to the body’s oestrogen receptors differently, and function differently. Activation of some receptors seems to promote cell growth, but isoflavones more often bind to oestrogen receptors with other effects, potentially acting as a tumour suppressor [71].

Different kinds of oestrogen receptors are present in different parts of the body. Activation of some receptors seems to promote cell growth. But studies suggest that isoflavones more often bind to oestrogen receptors with other effects, potentially acting as a tumour suppressor. Nevertheless, in Asian immigrants living in Western nations, whose diet includes more proteins and lipids and less fibers and soy, the risks for hormone-dependent cancers reach the same levels as the western population [72].

The main groups of phytoestrogens are lignans, coumestans, stilbenes and isoflavones.

Lignans, as components of plant cell walls, are found in many fiber-rich foods such as seeds (flax, pumpkin, sunflower, and sesame), whole grains (such as rye, oat, and barley), bran (such as wheat, oat, and rye), beans, fruits (especially berries), and cruciferous vegetables such as broccoli and cabbage [73].

The richest dietary source of plant lignans is flaxseed (*Linum usatissimum*), and crushing or milling flaxseed can increase lignan bioavailability [74].

Compared to isoflavones and lignans, coumestans are less prevalent in the human diet. Coumestans are primarily found in legume shoots and sprouts, primarily in clover and alfalfa, though small amounts have also been found in spinach and brussel sprouts [75]. Coumestrol is also found in trace levels in a variety of legumes, including split peas, pinto beans, lima beans, and soybean sprouts [75].

The most prevalent and studied stilbene, resveratrol, may be found in a number of plants and acts as a phytoalexin to ward off fungus infections. The skin of grapes (Vitis vinifera), red wine, and other highly pigmented fruit juices are the most recognized sources of resveratrol. Resveratrol is also present in pistachios, notably the papery skin surrounding the nut, and peanuts (Arachis). While flavonoids and resveratrol both have vascular effects that are frequently addressed, only the trans isomers of resveratrol have been found to have some phytoestrogenic effects [76].

Isoflavones are present in berries, wine, grains, and nuts, but are most abundant in soybeans, soy products, and other legumes [67,68].

Phytoestrogens, particularly isoflavones, exhibit both agonistic and antagonistic effects on ERβ and ERα receptors, depending on their concentration and affinity for various estrogen receptors [77]. This mechanism explains why phytoestrogens have a dual impact in ER-positive breast cancer cells, stimulating growth at low doses while inhibiting development at higher concentrations [78]. Coumestrol, genistein, and equol have a stronger affinity for ERβ [79,80].

Overall, phytoestrogens and their analogs inhibit cell cycle progression across different breast carcinomas by reducing mRNA or protein expression levels of cyclin (D1, E) and CDK (1, 2, 4, 6) and enhancing their inhibitors (p21, p27, p57) and tumor suppressor genes (APC, ATM, PTEN, SERPINB5) [73]. Even isoflavones, lignans, and resveratrol analogs influence cell cycle regulator expression, impacting different kinds of BC cell lines in vitro [81].

They also suppress the expression of oncogenic cyclin D1, as well as raise the levels of a variety of cyclin-dependent kinase inhibitors (p21, p27, and p57). Phytoestrogens, analogues, and derivatives may potentially influence BC behaviour, by interfering with estrogen production and metabolism as well as showing antiangiogenic, antimetastatic, and epigenetic effects. Furthermore, these bioactive molecules have the potential to reverse multi-drug resistance [81]. The benefits of phytoestrogens on human health, and particularly in BC patients, may also depend on their metabolism affected by the host’s microbiota present in the small and large intestine. For instance, genistein, equol, enterolignans, urolithins and other metabolites with higher binding affinity for estrogen receptors are more likely to yield beneficial effects.

Despite several research, the topic of whether phytoestrogens are useful or hurtful to people with BC remains unanswered: The answers are challenging and may vary with age, health state, and even gut microbial composition [82] (Table 2).

## 3. Estrobolome

The gut microbiota regulates the levels and bioavailability of estrogens, steroid hormones, and cytokines [94], all of which have a role in the development, progression, and outcome of the majority of BCs [95,96,97,98]. In addition to steroid hormones, BC may be influenced by adipose tissue hormones such as leptin and insulin, which are also regulated by intestinal microbiota.

Two main pathways have been identified through which microbiome influences the sexual hormonal balance. In the deconjugation-independent pathway, some phytoestrogens contained in food, such as plant lignans, are metabolized by specific intestinal bacteria into bioactive compounds. In the deconjugation-dependent pathway, several *genera* such as *Collinsella*, *Edwardsiella*, *Alistipes*, *Bacteroides*, *Bifidobacterium*, *Citrobacter*, *Clostridium*, *Dermabacter*, *Escherichia*, *Faecalibacterium*, *Lactobacillus*, *Marvinbryantia*, *Propionibacterium*, *Roseburia*, *Tannerella*, constituting the so called “estrobolome”, by the means of hydrolytic enzymes such as β-glucuronidases and β-glucosidases, can deconjugate estrogens excreted by the liver into the intestinal lumen as well as endocrine active food components, increasing their reabsorption through the entero-hepatic circulation [22,83,84,85]. Progesterone and testosterone bioavailability can also be affected by the sulfatase activity of certain gut microrganisms which convert circulating steroids into active hormones [85,86,87].

In premenopausal women following a “Western diet”, estrogen levels were found to be three times higher in feces and 15% to 20% lower in serum compared to a population of vegetarians eating a high fiber, moderate fat diet [99]. In another research, Asian immigrants showed 30% lower systemic estrogen levels compared to a similar population of American women consuming a diet higher in fats [100], probably due to the estrobolome composition, even though other variables such as lifestyle and oral supplements may also play a role [101]. Changes in the estrobolome composition induced by diet, physical activity, antibiotics and chemotherapeutics affect the systemic levels of estrogen and its metabolites through the entero-hepatic circulation [102] and this mechanism has been related to cancer progression in hormonal dependent BC patients and survivors [86,103,104].

Several studies showed that cancerogenesis can also be promoted by enhanced local exposures of breast tissue to hormonal triggers, both from estrogen and progesterone metabolites: an abundance of β-glucuronidase signalling has been found in nipple aspirate fluid of BC survivors [105], while BC tissue shows higher concentrations of estrogen metabolites compared to normal breast tissue [100,106]. Among the possible mechanisms leading to an increased production of progesterone metabolites in tumour microenvironment, *Bacillus cereus* seem to play a role in promoting cancer cells proliferation [107,108,109,110]. Among the gram-negative family of *Sphingomonadaceae*, *Sphyngomonas yanoikuyae*, relatively enriched in paired normal breast tissue as compared to cancer tissue [111], has shown the ability to digest monocyclic compounds and degrade estrogens within the breast tissue [112], which could interfere with cancerogenesis through the local estrogen bioavailability.

## 4. Interplay between Human Microbiota, Endocrine Disruptors, and Phytoestrogens

The complex relationship between microbiota and endocrine active compounds derived from diet act in a bidirectional way: enteric commensals can metabolize EDs into biologically active or inactive forms, while EDs may selectively induce the growth of specific bacterial populations.

The biotransformation of lignans is an intriguing example of how deeply the microbiota affect the metabolism of some xenobiotics: for instance, anhydrosecoisolariciresinol is converted by the gram-positive *Clostridium methoxybenzovorans* [60,61], the secoisolariciresinol diglucoside by *B. pseudocatenulatum* WC 401 and other *Bifidobacterium* strains through deglucosylation [62]. Among prenylflavonoids, a subgroup of chalcones and flavanones, the most significant are xanthohumol (XN) and desmethyxanthohumol (DMX) derived from hops, which are widely used in beer industry [113]: XN’s metabolite 8-prenylnaringenin (8-PN), produced in the gut by the commensal [114], is one of the most potent phytoestrogens [88], with a noticeable affinity for the ERα receptor [115,116].

These dietary-induced interactions between gut microbiota and hormonal balance may lead to a dysbiosis, thus affecting human health and diseases [117].

## 5. Role of the Endocrine Disruptors on Microbiota Composition

Several studies underline the association between EDs exposure and metabolic disorders, diabetes, obesity, and some neurobehavioral disorders [118], which have been related to gut dysbiosis, suggesting a role of gut microbiome and its products (post-biotics) as mediators of the effects induced by EDs in human metabolism [65].

Both the exposure to EDs and their bioactive metabolites may disrupt the microbiota composition and lead to dysbiosis [66], but also alter the microbiome functions and metabolic activities [119]. According to data from animal models, changes in the gut microbiota may have an im-pact on the host’s hepatic enzyme levels in addition to the levels of microbial enzymes [120].

Several EDs have been proved to promote dysbiosis or avoid bacterial growth both in vitro and in vivo [65], suggesting a significant influence on gut colonization with a consequence on host health. Furthermore, a “leaky gut” wall facilitates circulating EDs to flow into the intestinal milieu directly and, interacting with the enteric nervous system, could impact the composition and functions of the gut microbiota [121,122,123].

Clavel et al. showed that the isoflavone daidzein and its metabolites modulate the composition of gut microbiota in postmenopausal women after two months supplementation, finding an association between the equol production and the increase of the *F. prausnitzii* and *Lactobacillus-Enterococcus* groups [124]. In a long-term study exploring the effects of isoflavones supplementation on the faecal microbiota of healthy menopausal women, a significant change of microbial populations was recorded, but without any difference between equol-producers and non-producers [63].

In another study a 4 weeks supplementation with a pomegranate extract, ellagitannin and its metabolites reveal changes in the composition of gut microbiota (*Actinobacter*, *Firmicutes*, and *Verrucomicrobia*) on healthy subjects [125].

Luo et al. investigated the in vivo anti-obesity effect of flaxseed gums (FG) in obese rats and found the FG diet decreased the relative abundance of *Clostridiales* and increased the *Clostridium*, *Sutterella*, *Veillonella*, *Burkholderiales* and *Enterobacteriaceae* family in their gut microbiota [126]. The supplementation with syringaresinol, a plant lignan, increases the *Firmicutes/Bacteroidetes* ratio in an aging mouse model [64].

The resveratrol mechanisms of action are largely attributed to the modulation of gut microbiota and its metabolites. An in vitro study demonstrated a different conversion of trans-resveratrol into dihydroresveratrol, 3,4′–dihydroxybibenzyl, also known as lunularin, and 3,4′-dihydroxy-trans-stilbene, depending on the bacterial diversity of each individual’s faecal samples [89]. Chen et al. (2016) observed that modulation of gut microbiota induced by resveratrol reduced the levels of trimethylamine-N-oxide (TMAO) by inhibiting microbial trimethylamine (TMA) production and increased hepatic bile acid (BA) de novo synthesis [90]. An increase in *Bacteroides*/*Firmicutes* ratio was also observed in vivo after resveratrol supplementation in animal studies along with other effects, such as anti-diabetic effect [91], improved carbohydrate metabolism [92] and glucose homeostasis [93]. Giuliani et al., using an advanced gastrointestinal stimulator, showed that an extract containing a combination of t-resveratrol and ε-vinifrin induced changes in microbial functions and composition together with a strong decrease in the levels of SCFA and NH_4_^+^ [127].

## 6. Different Metabolic Pathways of Endocrine Disruptors Depending on Gut Microbiota

Gut microbiota are crucial in the conversion of EDs and phytoestrogens, such as isoflavones, ellagitannins, and lignans, into compounds with biological activity (equol, urolithins, and enterolignans, respectively) [61,128].

The enzymatic degradation of plant lignans, such as secoisolariciresinol, into phytoestrogens enterodiol and enterolactone by various gut bacteria, such as *Eggerthella lenta* and *Peptostreptococcus productus*, provides a model for the deconjugation-independent process [124]. Enterodiol and enterolactone may serve as selective estrogen modulators with anticancer properties [19,129] and a favorable prognostic impact in postmenopausal BC patients [130].

Van de Wiele et al. [131] reported that colonic microbiota can metabolize polyaromatic hydrocarbons into 1-hydroxy pyrene and 7-hydroxybenzo[a]pyrene, biologically active estrogen metabolites.

A recent review of Velmurugan et al. [66] focused on the role of gut microbiota in glucose dysregulation, glucose intolerance and insulin resistance induced by several classes of EDs from plastics, pesticides, synthetic fertilizers, electronic waste and food additives. They included bisphenols, dioxins, phthalates, organochlorines, organophosphates, fungicides, polychlorinated biphenyls and polychlorinated dibenzofurans, and other waste pollutants.

On the other hand, hyperglycemia induces changes of microbiota composition, favoring the growth of non-commensal germs, at the expense of beneficial phyla such as *Bacilli* (e.g., *Lactobacillus*), *Bacteroidetes*, *Proteobacteria* and *Actinobacteria* [66]. *Lactobacilli* can reduce pesticide toxicity and protect against EDs-induced oxidative stress by limiting contaminant absorption in the gut, strengthening tight junctions in the intestinal barrier, and activating host immunity [132]. Exposure to EDs, such as polychlorinated biphenyls, may impair intestinal permeability by suppressing the expression of tight junction proteins [133,134].

Gut dysbiosis is linked with many disorders such as obesity, diabetes, endocrine and immunological diseases [117,135,136,137,138,139,140], which have been proven as risk factors for BC in both pre- and post-menopausal women [141,142].

Furthermore, all major classes of EDs (bisphenols, phthalates, polychlorinated biphenyls, organochlorine pesticides, dioxins, and parabens) may increase the risk of obesity, developing insulin resistance and diabetes [143] by enhancing adipogenesis via hormone regulation of food intake, appetite, and disruption of pancreatic β-cell function [144,145,146,147,148,149]. Even the fungicide tributyltin, which has been shown to reduce gut microbial richness and microbiome composition in mice [150], stimulates adipogenesis by interacting with nuclear PPAR γ and its heteromeric companion retinoid X receptor.

The interplay between EDs and human microbiota affects BC risk and clinical management not only through the sexual hormonal balance, but also through the innate as well as the acquired immunity [132,151,152,153,154], but these fundamental pathways are beyond the topic of the present review.

Beside this, a plethora of studies show that the gut microbiome affects the side effects, the toxicity and the outcomes of anticancer treatments such as radiation therapy, chemotherapy, immunotherapy and hormone therapy.

On the other hand, anticancer agents such as letrozole, an aromatase inhibitor, are associated with a time-dependent reduction of phylogenetic richness in the gut microbiota and a significant decrease in overall species [107].

Based on these results, the gut microbiota could become a key part of a microbiota-host-cancer triad as a new paradigm in order to better predict patients’ response to therapies and build a more tailored approach to cancer patients.

## 7. Conclusions

Endocrine disruptors and phytoestrogens interact with the human microbiota both at the intestinal and the breast tissue levels, affecting estrogens’ balance, bioavailability, and functions. This complex interplay results in a modification of BC cells behaviours, at least for hormonal dependent tumors, which account for more than 70% of cases globally.

A better understanding of this interplay, as well as the chance of modulating the exposure to EDs and targeting the microbiome composition (via dietary interventions and probiotics) could pave the way to a new oncobiotic approach in order to improve the clinical management of BC patients.

## Figures and Tables

**Table 1 jcm-12-03158-t001:** Interplay between endocrine disruptors and microbiota with potential drivers of breast cancer.

Source	Molecules	Microrganisms	Outcome	References
Foods	Lignans Isoflavones	*C. methoxybenzovorans* *B. pseudocatenulatum* WC 401 *Firmicutes* *Bacteroidetes* *F. prausnitzii* *Lactobacillus* *Enterococcus*	Estrogen Bioavailability	[60,61,62,63,64]
Plastics	BPA BPS	*Helicobacteraceae* *Firmicutes* *Clostridia*	Lipogenesis Gluconeogenesis Tumor proliferation Metastatic spread	[58,59,65,66]
Pesticides	Organophosphates DDT DDE PCB	*Bacteroides*, *Burkholderiales* *Clostridiaceae* *Erysiopelotrichaceae* *Coprobacillus* *Lachnospiraceae* *Staphylococcaceae*	Gluconeogenesis Oxidative stress Changes in insulin and ghrelin secretion	[49,50,65,66]
Heavy metals	Arsenic Lead Cadmium	*Bacteroides* *Firmicutes* *Proteobacteria*	Altered gluconeogenesis Lipogenesis Inflammation Body fat	[65,66]

BPA, Bisphenol A; BPS, Bisphenol S; DDT, dichloro-diphenyl-trichloroethane; DDE, Dichloro-diphenyl-dichloroethylene; PCB, polychlorinated biphenyl.

**Table 2 jcm-12-03158-t002:** Interplay between phytoestrogens and their metabolites with microrganism.

Chemical Family	Molecules	Microrganisms	References
Lignans	Anhydrosecoisolariciresinol Secoisolariciresinol diglucoside Syringaresinol	*C. methoxybenzovorans* *B. pseudocatenulatum* WC 401 *Firmicutes* *Bacteroidetes*	[60,61,62,64]
Isoflavones	Coumestrol Genistein Equol Daidzein	*F. prausnitzii* *Lactobacillus* *Enterococcus*	[63]
Steroids	Estradiol Estrone	*Collinsella*, *Edwardsiella*, *Alistipes*, *Bacteroides*, *Bifidobacterium*, *Citrobacter*, *Clostridium*, *Dermabacter*, *Escherichia*, *Faecalibacterium*, *Lactobacillus*, *Marvinbryantia*, *Propionibacterium*, *Roseburia*, *Tannerella*	[22,83,84,85,86,87]
Prenylflavonoids	Xanthohumol Desmethyxanthohumol	*E. limosum*	[88]
Stilbenes	Resveratrol Trans-resveratrol Dihydroresveratrol 3,4′–dihydroxybibenzyl, 3,4′-dihydroxy-trans-stilbene	*Firmicutes* *Bacteroidetes*, *Actinobacteria* *Verrucomicrobia*, *Cyanobacteria*	[89,90,91,92,93]

## Data Availability

The data are available upon request from the corresponding author.

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
