# Peer review of "Endocrine Disruptors in Food, Estrobolome and Breast Cancer"

_jcm, 2023, doi:10.3390/jcm12093158_

Round 1
Reviewer 1 Report (New Reviewer)
Line 21 ,,As regards”-Change it to “Regarding”
Line 35- Throughout this manuscript there are several pharagraphs consisting of just one sentence. Paragraphs need to contain more than one sentence to stand alone. Suggest combining single sentence paragraphs or otherwise modifying.
Line 73 ,,As regards”-Change it to “Regarding”
Line 90 ,,dioxins”-To my knowledge, dioxins are not used as solvents or lubricants.Suggest removing
Line 126- remove the
Line 135 “BPS”- Define BPS
Line 145 Lower- Confirm that Asian populations have lower rates of hormone dependent cancers. With the increase of these hormones in diet, one would suspect the opposite trend. If the statement is true, please give a brief explanation as to why this is true.
Line 162-Additionally, it has been noted that after an insect and fungal attack, the coumestrol levels in legumes rise. [68].- This is irrelevant. suggest removing.
Line 212- “regulates”-remove the “s”
Line 230 –“ 5a-pregnanes (5aP)”-Why are these levels higher?Is this significant?
Line 250-phytoestrogen-Add an “s” to phytoestrogen.
Line 266-let-this word should be plural.
Line 302-enzimatic-change to enzymatic
Line 329 “ act as”-replace with be
Line 342 “Pharmacomicrobiomics is defined as the effect of microbiome variations on drug absorption, distribution, metabolism, excretion, toxicity and overall response [95;156-158]”. – This is out of context. Suggest moving to the introduction.

Author Response
Dear Mr. Dragisa Medojevic,
Thank you for giving me the opportunity to submit a revised draft of my manuscript titled “Endocrine disruptors in food, estrobolome and breast cancer” to Journal of Clinical Medicine.
We appreciate the time and effort that you and the reviewers have dedicated to providing your valuable feedback on our manuscript.
We are grateful to the reviewers for their insightful comments on our paper.
We have been able to incorporate changes to reflect most of the suggestions provided by the reviewers.
Here is a response to the reviewers’ comments and concerns:
Thank you for pointing this out we have done the correction on spelling and grammatical errors according to your suggestion in lines 21, 35, 73, 90, 126, 162, 212,230,250 266, 302, 329 and 342.
Regarding the line 145 we agree with this and have incorporated your suggestion throughout the manuscript providing to add the explanation requested.
We look forward to hearing from you in due time regarding our submission and to respond to any further questions and comments you may have.
Sincerely,
Alessio Filippone
Reviewer 2 Report (New Reviewer)
Filippone et al present in a comprehensive way the interaction between endocrine disruptors, phytoestrogens, gut microbiota, and breast cancer. Although the review is lengthy and detailed, there is a lack of emphasis on the data-evidence that links the endocrine disruptors- estrobolome with the development or progress of breast cancer.
I would suggest that the authors should concentrate more on this part of the review and a) expand on the studies linking endocrine disruptors to breast cancer b) discuss more extensively the studies referenced in lines 223-224
The manuscript would also benefit from a table presenting the aforementioned studies and the evidence that they provide towards linking endocrine disruptors and estrobolome with breast cancer.
Author Response
Dear Mr. Dragisa Medojevic,
Thank you for giving me the opportunity to submit a revised draft of my manuscript titled “Endocrine disruptors in food, estrobolome and breast cancer” to Journal of Clinical Medicine.
We appreciate the time and effort that you and the reviewers have dedicated to providing your valuable feedback on our manuscript.
We are grateful to the reviewers for their insightful comments on our paper.
We have been able to incorporate changes to reflect most of the suggestions provided by the reviewers.
Here is a response to the reviewers’ comments and concerns:
Thank you for pointing this out we have done the changes according to your suggestion providing to add the explanation requested in point A and B.
Regarding the last suggestions we agree with this and have incorporated a further table throughout the manuscript. This table regard the interplay between endocrine disruptors and microbiota with potential drivers of breast cancer.
We look forward to hearing from you in due time regarding our submission and to respond to any further questions and comments you may have.
Sincerely,
Alessio Filippone
This manuscript is a resubmission of an earlier submission. The following is a list of the peer review reports and author responses from that submission.
Round 1
Reviewer 1 Report
Dear authors,
This is an important review article regarding a hot topic in breast cancer and its relationship with the gut and local microbiota (“breast-gut axis”).
However, several issues must be addressed or even considered before the potential acceptance for publication:
§ Mini-review article; and with no methods discriminated… it is a narrative review (not a scoping or systematic review)
§ Complete the breast cancer subtypes description.
§ https://www.annalsofoncology.org/article/S0923-7534(19)31287-6/pdf (table 2)
§ May consider a “very new” subtype with an emerging targeted therapy like trastuzumab deruxtecan The “her2-low disease”
§ Tarantino P, Hamilton E, Tolaney SM, Cortes J, Morganti S, Ferraro E, Marra A, Viale G, Trapani D, Cardoso F, Penault-Llorca F, Viale G, Andrè F, Curigliano G. HER2-Low Breast Cancer: Pathological and Clinical Landscape. J Clin Oncol. 2020 Jun 10;38(17):1951-1962. doi: 10.1200/JCO.19.02488. Epub 2020 Apr 24. PMID: 32330069.
§ Explore more the phytoestrogens metabolism (e.g., isoflavones and lignans) and its conversion to active metabolites that can protect against breast cancer. Interventional studies have shown that the ability of some individuals’ intestinal microbiota to convert isoflavones and lignans into equol and enterolignans, respectively, may result in a reduced risk of hormone-dependent diseases, such as BC. Differences between occidental and Asiatic populations.
§ Yuan, J. P., Wang, J. H., and Liu, X. (2007). Metabolism of dietary soy isoflavones to equol by human intestinal microflora–implications for health. Mol. Nutr. Food Res. 51, 765–781. doi: 10.1002/mnfr.200600262 ;
§ Gaya, P., Medina, M., Sánchez-Jiménez, A., and Landete, J. M. (2016). Phytoestrogen Metabolism by Adult Human Gut Microbiota. Molecules 21:1034. doi: 10.3390/molecules21081034
§ Poluzzi, E., Piccinni, C., Raschi, E., Rampa, A., Recanatini, M., and De Ponti, F. (2014). Phytoestrogens in postmenopause: the state of the art from a chemical, pharmacological and regulatory perspective. Curr. Med. Chem. 21, 417–436. doi: 10.2174/09298673113206660297
§ Alpuim Costa D, Nobre JG, Batista MV, Ribeiro C, Calle C, Cortes A, Marhold M, Negreiros I, Borralho P, Brito M, Cortes J, Braga SA, Costa L. Human Microbiota and Breast Cancer-Is There Any Relevant Link?-A Literature Review and New Horizons Toward Personalised Medicine. Front Microbiol. 2021 Feb 25;12:584332. doi: 10.3389/fmicb.2021.584332. PMID: 33716996; PMCID: PMC7947609.
§ Explore other bioactive compounds that modulate breast cancer risk, such as short-chain fatty acids, lithocholic acid and cadaverine.
§ Kovács, T., Mikó, E., Vida, A., Sebõ, É, Toth, J., et al. (2019). Cadaverine, a metabolite of the microbiome, reduces breast cancer aggressiveness through trace amino acid receptors. Sci. Rep. 9:1300. doi: 10.1038/s41598-018-37664-7
§ Alpuim Costa D, Nobre JG, Batista MV, Ribeiro C, Calle C, Cortes A, Marhold M, Negreiros I, Borralho P, Brito M, Cortes J, Braga SA, Costa L. Human Microbiota and Breast Cancer-Is There Any Relevant Link?-A Literature Review and New Horizons Toward Personalised Medicine. Front Microbiol. 2021 Feb 25;12:584332. doi: 10.3389/fmicb.2021.584332. PMID: 33716996; PMCID: PMC7947609.
§ Mikó, E., Vida, A., Kovács, T., Ujlaki, G., Trencsényi, G., et al. (2018). Lithocholic acid, a bacterial metabolite reduces breast cancer cell proliferation and aggressiveness. Biochim. Biophys. Acta Bioenerg. 1859, 958–974. doi: 10.1016/ j.bbabio.2018.04.002
§ Include a table and/or figure to better capture our attention (studies data, the “breast-gut axis”, etc.)
§ Try to consider the pharmacomicrobiomics regarding breast cancer and gut/local microbiota
§ Alpuim Costa D, Nobre JG, Batista MV, Ribeiro C, Calle C, Cortes A, Marhold M, Negreiros I, Borralho P, Brito M, Cortes J, Braga SA, Costa L. Human Microbiota and Breast Cancer-Is There Any Relevant Link?-A Literature Review and New Horizons Toward Personalised Medicine. Front Microbiol. 2021 Feb 25;12:584332. doi: 10.3389/fmicb.2021.584332. PMID: 33716996; PMCID: PMC7947609.
§ Vitorino M, Baptista de Almeida S, Alpuim Costa D, Faria A, Calhau C, Azambuja Braga S. Human Microbiota and Immunotherapy in Breast Cancer - A Review of Recent Developments. Front Oncol. 2022 Jan 28;11:815772. doi: 10.3389/fonc.2021.815772. PMID: 35155205; PMCID: PMC8832278.
§ Barroso-Sousa, R., Ajami, N., Keenan, T. E., Andrews, C., Pittenger, J. L., Wulf, G., et al. (2020). SABC 2019 Abstract P3-09-16: Fecal microbiome and association with outcomes among patients (pts) receiving eribulin (E) +/- pembrolizumab (P) for hormone receptor positive (HR+) metastatic breast cancer (MBC). SABC 80, 3–09–16. doi: 10.1158/1538-7445.SABCS19-P3-09
§ Panebianco, C., Andriulli, A., and Pazienza, V. (2018). Pharmacomicrobiomics: exploiting the drug-microbiota interactions in anticancer therapies. Microbiome 6:92. doi: 10.1186/s40168-018-0483-7
Thank you.
Author Response
Dear Lorina Seras,
Thank you for giving me the opportunity to submit a revised draft of our manuscript titled "Endocrine disruptors in food, estrobolome and breast cancer" to JCM.
We appreciate the time and effort that you and the reviewers have dedicated to providing your valuable feedback on our manuscript. We are grateful to the reviewers for their insightful comments on our paper.
We have been able to incorporate changes to reflect most of the suggestions provided by the reviewers. You can find the changes within the manuscript using the tool revision.
Here is a point-by-point response to the reviewers’ comments and concerns:
Comment: Mini-review article; and with no methods discriminated… it is a narrative review (not a scoping or systematic review)
Response: Thank you for the comment. We agree with this comment. Therefore, we have add and updated more evidences about the review.
Comment: Complete the breast cancer subtypes description. https://www.annalsofoncology.org/article/S0923-7534(19)31287-6/pdf (table 2). May consider a “very new” subtype with an emerging targeted therapy like trastuzumab deruxtecan The “her2-low disease”
Tarantino P, Hamilton E, Tolaney SM, Cortes J, Morganti S, Ferraro E, Marra A, Viale G, Trapani D, Cardoso F, Penault-Llorca F, Viale G, Andrè F, Curigliano G. HER2-Low Breast Cancer: Pathological and Clinical Landscape. J Clin Oncol. 2020 Jun 10;38(17):1951-1962. doi: 10.1200/JCO.19.02488. Epub 2020 Apr 24. PMID: 32330069.
Response: Thank you for the suggestion. We have updated the classification citing the suggested paper.
Comment: Explore more the phytoestrogens metabolism (e.g., isoflavones and lignans) and its conversion to active metabolites that can protect against breast cancer. Interventional studies have shown that the ability of some individuals’ intestinal microbiota to convert isoflavones and lignans into equol and enterolignans, respectively, may result in a reduced risk of hormone-dependent diseases, such as BC. Differences between occidental and Asiatic populations.
- Yuan, J. P., Wang, J. H., and Liu, X. (2007). Metabolism of dietary soy isoflavones to equol by human intestinal microflora–implications for health. Mol. Nutr. Food Res. 51, 765–781. doi: 10.1002/mnfr.200600262 ;
- Gaya, P., Medina, M., Sánchez-Jiménez, A., and Landete, J. M. (2016). Phytoestrogen Metabolism by Adult Human Gut Microbiota. Molecules 21:1034. doi: 10.3390/molecules21081034
- Poluzzi, E., Piccinni, C., Raschi, E., Rampa, A., Recanatini, M., and De Ponti, F. (2014). Phytoestrogens in postmenopause: the state of the art from a chemical, pharmacological and regulatory perspective. Curr. Med. Chem. 21, 417–436. doi: 10.2174/09298673113206660297
- Alpuim Costa D, Nobre JG, Batista MV, Ribeiro C, Calle C, Cortes A, Marhold M, Negreiros I, Borralho P, Brito M, Cortes J, Braga SA, Costa L. Human Microbiota and Breast Cancer-Is There Any Relevant Link?-A Literature Review and New Horizons Toward Personalised Medicine. Front Microbiol. 2021 Feb 25;12:584332. doi: 10.3389/fmicb.2021.584332. PMID: 33716996; PMCID: PMC7947609.
Response: Thak you for the suggestion. We had provide to describe all the suggestion including isoflavones, lignans and other type of phytoestrogen by clinical and population studies citing the paper suggested.
Comment: Explore other bioactive compounds that modulate breast cancer risk, such as short-chain fatty acids, lithocholic acid and cadaverine.
- Kovács, T., Mikó, E., Vida, A., Sebõ, É, Toth, J., et al. (2019). Cadaverine, a metabolite of the microbiome, reduces breast cancer aggressiveness through trace amino acid receptors. Sci. Rep. 9:1300. doi: 10.1038/s41598-018-37664-7 Alpuim Costa D, Nobre JG, Batista MV, Ribeiro C, Calle C, Cortes A, Marhold M, Negreiros I, Borralho P, Brito M, Cortes J, Braga SA, Costa L. Human Microbiota and Breast Cancer-Is There Any Relevant Link?-A Literature Review and New Horizons Toward Personalised Medicine. Front Microbiol. 2021 Feb 25;12:584332. doi: 10.3389/fmicb.2021.584332. PMID: 33716996; PMCID: PMC7947609.
- Mikó, E., Vida, A., Kovács, T., Ujlaki, G., Trencsényi, G., et al. (2018). Lithocholic acid, a bacterial metabolite reduces breast cancer cell proliferation and aggressiveness. Biochim. Biophys. Acta Bioenerg. 1859, 958–974. doi: 10.1016/ j.bbabio.2018.04.002
Response: Thak you for the suggestion. We had provide to describe all the suggestion citing the paper mentioned.
Comment: Include a table and/or figure to better capture our attention (studies data, the “breast-gut axis”, etc.)
Response: We agree with this and have incorporated your suggestion throughout the manuscript. We have summerize the interplay between phytoestrogens and their metabolites with microrganism in a table.
Comment: Try to consider the pharmacomicrobiomics regarding breast cancer and gut/local microbiota
- Alpuim Costa D, Nobre JG, Batista MV, Ribeiro C, Calle C, Cortes A, Marhold M, Negreiros I, Borralho P, Brito M, Cortes J, Braga SA, Costa L. Human Microbiota and Breast Cancer-Is There Any Relevant Link?-A Literature Review and New Horizons Toward Personalised Medicine. Front Microbiol. 2021 Feb 25;12:584332. doi: 10.3389/fmicb.2021.584332. PMID: 33716996; PMCID: PMC7947609.
- Vitorino M, Baptista de Almeida S, Alpuim Costa D, Faria A, Calhau C, Azambuja Braga S. Human Microbiota and Immunotherapy in Breast Cancer - A Review of Recent Developments. Front Oncol. 2022 Jan 28;11:815772. doi: 10.3389/fonc.2021.815772. PMID: 35155205; PMCID: PMC8832278.
- Barroso-Sousa, R., Ajami, N., Keenan, T. E., Andrews, C., Pittenger, J. L., Wulf, G., et al. (2020). SABC 2019 Abstract P3-09-16: Fecal microbiome and association with outcomes among patients (pts) receiving eribulin (E) +/- pembrolizumab (P) for hormone receptor positive (HR+) metastatic breast cancer (MBC). SABC 80, 3–09–16. doi: 10.1158/1538-7445.SABCS19-P3-09
- Panebianco, C., Andriulli, A., and Pazienza, V. (2018). Pharmacomicrobiomics: exploiting the drug-microbiota interactions in anticancer therapies. Microbiome 6:92. doi: 10.1186/s40168-018-0483-7
Response: We agree with this and have incorporated your suggestion throughout the manuscript citing the paper suggested.
We look forward to hearing from you regarding our submission and to respond to any further questions and comments you may have.
Sincerely
Alessio Filippone
